# Tendinopathies and Pain Sensitisation: A Meta-Analysis with Meta-Regression

**DOI:** 10.3390/biomedicines10071749

**Published:** 2022-07-20

**Authors:** Davide Previtali, Alberto Mameli, Stefano Zaffagnini, Paolo Marchettini, Christian Candrian, Giuseppe Filardo

**Affiliations:** 1Service of Orthopaedics and Traumatology, Department of Surgery, EOC, 6900 Lugano, Switzerland; alberto_mameli@icloud.com (A.M.); christian.candrian@eoc.ch (C.C.); ortho@gfilardo.com (G.F.); 2II Clinica Ortopedica e Traumatologica, IRCCS Istituto Ortopedico Rizzoli, 40136 Bologna, Italy; stefano.zaffagnini@unibo.it; 3Fisiopatologia e Terapia del Dolore, Dipartimento di Farmacologia, Careggi Università di Firenze, 50134 Firenze, Italy; marchettini.paolo@hsr.it; 4Terapia del Dolore, Centro Diagnostico Italiano, 20147 Milan, Italy; 5Faculty of Biomedical Sciences, Università della Svizzera Italiana, 6900 Lugano, Switzerland; 6Applied and Translational Research Center, IRCCS Istituto Ortopedico Rizzoli, 40136 Bologna, Italy

**Keywords:** tendinopathy, pain, pain sensitisation

## Abstract

The presence of pain sensitisation has been documented and reported as being a possible cause of treatment failure and pain chronicity in several musculoskeletal conditions, such as tendinopathies. The aim of the present study is to analyse existing evidence on pain sensitisation in tendinopathies comparing the local and distant pain thresholds of healthy and affected subjects with distinct analysis for different tendinopathies. PubMed, Cochrane Central Register, Scopus, and Web Of Science were systematically searched after registration on PROSPERO (CRD42020164124). Level I to level IV studies evaluating the presence of pain sensitisation in patients with symptomatic tendinopathies, documented through a validated method, were included. A meta-analysis was performed to compare local, contralateral, and distant pain thresholds between patients and healthy controls with sub-analyses for different tendinopathies. Meta-regressions were conducted to evaluate the influence of age, activity level, and duration of symptoms on results. Thirty-four studies out of 2868 were included. The overall meta-analysis of local pressure pain thresholds (PPT) documented an increased sensitivity in affected subjects (*p* < 0.001). The analyses on contralateral PPTs (*p* < 0.001) and distant PPTs (*p* = 0.009) documented increased sensitivity in the affected group. The results of the sub-analyses on different tendinopathies were conflicting, except for those on lateral epicondylalgia. Patients’ activity level (*p* = 0.02) and age (*p* = 0.05) significantly influenced local PPT results. Tendinopathies are characterized by pain sensitisation, but, while features of both central and peripheral sensitisation can be constantly detected in lateral epicondylalgia, results on other tendinopathies were more conflicting. Patients’ characteristics are possible confounders that should be taken into account when addressing pain sensitisation.

## 1. Introduction

Tendinopathies are common injuries, frequently sport-related and affecting a young and active population [1,2,3,4]. The characteristics of the affected patients imply a relevant economic burden for society [5], which is also increased by the difficulties in properly addressing the disease with the available treatments, often leading to symptoms’ chronicity. As well as tenderness to palpation and impaired function, tendon pain with loading is the main symptom of tendinopathies but their origins are still not well understood, thus undermining the treatment possibilities [6,7].

The presence of pain sensitisation, characterized by increased (hyperalgesia) and abnormal (allodynia) pain perception, has been documented and reported as a possible cause of treatment failure and pain chronicity in several musculoskeletal conditions [8,9,10]. Patients with chronic tendinopathies present increased perceived pain during tendon palpation and report pain during normally pain-free movements (i.e., jumping, squatting, running, shoulder abduction), which have been recognized as possible manifestations of hyperalgesia and allodynia [6]. In this regard, quantitative sensory testing (QST) protocols, i.e., diagnostic tests able to evaluate pressure, cold, and heat pain thresholds, documented higher local pain sensitivity in subjects affected by chronic tendinopathies [11,12]. This has led to the identification of an abnormal pain processing pathway as an underlying mechanism that may explain some of the features of tendon pain [13]. Nonetheless, a recent study failed to demonstrate increased pain sensitivity in patellar and Achilles tendinopathies, suggesting that central sensitization may play only a minor role in these diseases [14]. The reasons for this discrepancy are still not completely understood, with authors suggesting the importance of the epidemiologic characteristics of the different tendinopathies: upper limb tendinopathies usually affect older and less active patients with a possible influence on quantitative sensory tests results [15]. Thus, the relevance and the impact of pain sensitisation on symptoms of patients affected by different chronic tendinopathies still need to be determined.

The aim of this meta-analysis was to investigate the evidence on pain sensitisation in chronic tendinopathies. In particular, a quantitative synthesis comparing local and distant pain threshold of healthy and affected subjects was performed, with distinct analysis for different chronic tendinopathies and a further analysis based on patient age, symptoms’ duration, and activity level.

## 2. Materials and Methods

### 2.1. Data Source

After the registration of the protocol on PROSPERO (CRD42020164124), PubMed (MEDLINE), the Cochrane Central Register of Controlled Trials (CENTRAL), Scopus, Web of Science, and gray literature were systematically searched on 1 March 2022.

The following string was used: (tendon OR tendinopathy OR tendinopathies OR epicondylalgia OR tennis elbow OR jumpers knee OR patellar tendon OR Achilles OR rotator cuff OR shoulder impingement) AND (pain) AND (sensitization OR hyperalgesia OR threshold OR hypersensitivity OR algometry OR allodynia OR quantitative sensory testing OR neuropathic). No Institutional Review Board permission was needed to retrieve data.

### 2.2. Study Selection

The study selection process was performed between 2 March and 14 March 2022. Firstly, duplicates were removed, and then all titles and abstracts were checked to retrieve all eligible articles. The full-text article was read if not enough information on eligibility could be obtained from the abstract. Inclusion criteria were: Level I to level IV studies; studies evaluating the presence of pain sensitisation through a validated method (questionnaires or QST) or comparing QST between healthy and affected subjects; studies including patients with symptomatic chronic (pain duration > 3 months) tendinopathies. No time or language limitations were set. The Preferred Reporting Items for Systematic Reviews and Meta-analysis (PRISMA) guidelines were used [16]. Two authors (DP, AM) independently performed the article selection process, with any disagreement on study eligibility solved by a third author (GF).

### 2.3. Data Extraction and Study Outcomes

The data extraction process was performed between 15 March and 10 April 2022. Information on methodology from all eligible studies included level of evidence, publication year, study design, technique of pain sensitisation assessment, type of tendinopathy evaluated and diagnostic technique, origin of patients, number of patients included, and follow-up length. Information from all eligible trials on the characteristics of the study population included: inclusion/exclusion criteria, sex, age, body mass index, activity level, comorbidities, PROMs, duration of symptoms, functional scores, presence of pain sensitisation (number of patients or prevalence), results of pain sensitisation assessment (questionnaires, QST), and the influence of pain sensitisation on the clinical outcome. Two review authors extracted the trials’ information independently and in duplicate using a standardized extraction form. The primary outcome was the difference in local pressure pain threshold (PPT) between patients with tendinopathies and healthy subjects. Contralateral and distant PPT, local heat pain threshold (HPT), and local cold pain threshold (CPT) were also evaluated. Separate analyses for different tendinopathies were conducted, with a sub-analysis based on the level of activity of the included patients. The influence of age and activity level of the patients on the results of the meta-analyses was evaluated through a meta-regression since they were identified in the literature as possible predictors of pain sensitisation [15]. An evaluation of the differences in terms of conditioned pain modulation and temporal summation was not possible due to the lack of data.

### 2.4. Quality Assessment

The quality of the included studies was assessed by two separate authors (DP, AM) using the Newcastle–Ottawa scale for case–control studies [17]. This uses a star-rating system to judge the study quality based on case selection (case definition, representativeness of cases, controls’ definition, and controls’ selection), comparability of cases and controls (in terms of age and other variables), and exposure (ascertainment of exposure, same method of ascertainment between cases and controls, and non-response rate). In the case of disagreement between the two authors, the studies were discussed and a consensus was reached.

### 2.5. Statistical Analysis

Continuous data were expressed as means and standard deviations and compared as standardized mean differences, whereas binary data were expressed as frequencies and compared as risk ratios. A meta-analysis was performed to compare local, contralateral, and distant PPTs, local HPTs, and local CPTs between patients with painful tendinopathies and healthy controls. Three separate linear meta-regressions were conducted to evaluate the influence of patient age, activity level, and duration of symptoms on the reported results. A multiple meta-regression was not feasible due to the low number of included studies [18]. Separate analyses for different tendinopathies were conducted. The random effect model with Knapp–Hartung–Sidik–Jonkman adjustment was used and results were expressed as standardized mean differences (SMD). The statistical analysis was performed with meta (v4.9-7, Schwarzer G, 2007) and metafor (v2.1-0, Viechtbauer, W., 2010) packages in RStudio (v1.2.5019; 250 Northern Ave., Boston, MA, USA, 02210).

## 3. Results

### 3.1. Characteristics of the Included Studies and Patients

This meta-analysis included 34 studies out of 2868 retrieved records (Figure 1) [11,12,19,20,21,22,23,24,25,26,27,28,29,30,31,32,33,34,35,36,37,38,39,40,41,42,43,44,45,46]. These 34 studies addressed 9 different tendinopathies: 11 were on lateral epicondylalgia, 5 were on Achilles tendinopathy, 4 were on shoulder impingement syndrome, 3 were on great trochanter pain syndrome, 3 were on patellar tendinopathy, 2 were on rotator cuff tears, 2 were on plantar heel pain, 2 were on all lower limb tendinopathies (greater trochanteric pain syndrome, quadriceps tendinopathy, patellar tendinopathy, Achilles tendinopathy, and plantar heel pain), 1 was on ilio-tibial band syndrome, and 1 included both patients with patellar and Achilles tendinopathy. Twenty-eight studies evaluated the differences between affected and healthy subjects in terms of pain thresholds, whereas 6 studies (2 on rotator cuff tears, 1 on greater trochanteric pain syndrome, 1 on Achilles tendinopathy, and 2 on lower limb tendinopathies) reported the prevalence of features of pain sensitisation in a cohort of patients using validated questionnaires (i.e., painDETECT, Dolour Neuropathic 4, Central Sensitization Index) or QST protocols (conditioned pain modulation).

A total of 1831 patients were included: 406 with lateral epicondylalgia, 338 with Achilles tendinopathy, 290 with great trochanter pain syndrome, 287 with plantar heel pain, 211 with rotator cuff tears, 188 with patellar tendinopathy, 87 with shoulder impingement syndrome, 9 with ilio-tibial band syndrome, and 4 with quadriceps tendinopathy. In the reported studies the mean age ranged from 21.9 to 65.7, the mean BMI ranged from 22.2 to 33.7, and the mean pain duration ranged from 3 months to 10.6 years. Details on the characteristics of the included studies and patients are reported in Table 1.

### 3.2. Local Pain Thresholds

The overall meta-analysis of local PPT documented an increased sensitivity in affected subjects (SMD: −1.54; 95% C.I. −1.92, −1.16; *p* < 0.001) (Figure 2). A decreased local PPT was found in 22 out 23 studies. Only Plinsinga et al. found a similar local PPT between healthy controls and patients with Achilles tendinopathy [14]. The sub-analysis, based on specific tendinopathy, documented a significant difference in local PPT only in patients suffering from epicondylalgia (SMD: −1.59; 95% C.I. −2.06, −1.12; *p* < 0.001) and greater trochanter pain syndrome (SMD: −1.49; 95% C.I. −1.94, −1.03; *p* < 0.001). No significant difference was documented in the sub-analyses on shoulder impingement syndrome, patellar tendinopathy, and plantar heel pain even though all single studies found an increased sensitivity in the affected patients. Similarly, no significant difference was found in the sub-analysis on Achilles tendinopathy, with three studies documenting an increased sensitivity in the affected patients and one study documenting a similar PPT between healthy and affected subjects.

The overall analysis on HPTs found no difference between healthy and affected subjects with only the study of Coombes et al. documenting an increased sensitivity in patients with epicondylalgia and the study of Eckenrode et al. documenting an increased sensitivity in patients with Achilles tendinopathy [22,23]. No significant difference was documented in all sub-analyses based on the specific tendinopathy (Figure 3). A significant difference (SMD: 0.47; 95% C.I. 0.21, 0.72; *p* < 0.001), indicating an increased sensitivity in affected subjects, was found in the meta-analysis evaluating CPTs with the studies of Coombes et al. and of Ruiz-Ruiz et al., both documenting an increased sensitivity to cold stimuli in subjects with epicondylalgia, and the study of Plinsinga et al. documenting an increased sensitivity in the subjects affected by greater trochanter pain syndrome [22,38,41]. The sub-analysis on lateral epicondylalgia and on greater trochanter pain syndrome found a significant difference between groups (SMD: 0.68; 95% C.I. 0.46, 0.90; *p* < 0.001; SMD: 0.48; 95% C.I. 0.07, 0.89; *p* < 0.001) (Figure 3).

### 3.3. Contralateral and Distant Pain Thresholds

The overall analysis on contralateral PPT documented an increased sensitivity in the affected group (SMD: −0.87; 95% C.I. −1.18, −0.56; *p* < 0.001) (Figure 4). Results of the single studies were heterogeneous with 11 studies finding a lower PPT in the affected subjects and 6 studies reporting no significant difference between groups. The sub-analyses including only patients suffering from epicondylalgia, plantar heel pain, and patellar tendinopathy documented significantly lower PPTs in patient groups (SMD: −0.88; 95% C.I. −1.25, −0.51; *p* < 0.001; SMD: −1.01; 95% C.I. −1.51, −0.52; *p* < 0.001; SMD: −1.93; 95% C.I. −2.55; −1.32; *p* < 0.001). No significant difference in contralateral PPTs was documented in the sub-analyses on shoulder impingement syndrome and Achilles tendinopathy, with all studies on these tendinopathies reporting no differences between groups except for the study of Vallance et al. on Achilles tendinopathy [44].

The overall analysis on distant PPTs reported a significantly increased sensitivity in patients with tendinopathies (SMD: −1.01; 95% C.I. −1.67, −0.38; *p* = 0.009) (Figure 5). In particular, 11 studies reported lower distant PPT and 8 studies documented no difference between groups. Only the sub-analysis on greater trochanteric pain syndrome found a significantly increased sensitivity in the affected patients (SMD: −0.58; 95% C.I. −0.99, −0.17; *p* = 0.01). None of the other sub-analyses based on a specific tendinopathy documented an increased pain sensitivity in the affected patients. In particular, an increased sensitivity was documented in two out of three studies on shoulder impingement syndrome, in five out of six studies on epicondylalgia, in one out of two of the studies on patellar tendinopathy, in one out of five studies on Achilles tendinopathy, and in one out of two studies on plantar heel pain.

### 3.4. Patient Characteristics Influencing Pain Sensitivity

The meta-regression showed a significant influence of patient activity level on local PPT, with an increased difference in pain threshold in studies on athletes (β = −1.1, *p* = 0.02, R^2^ 19.2%). A significant influence on the results was also found for patient age with less difference in local PPT between healthy and affected subjects in studies including older patients (β = 0.04, *p* = 0.05, R^2^ 12.9%). No influence on the results of the analysis on local PPT was documented for the duration of symptoms.

When the meta-regression was conducted on the results of the meta-analyses of contralateral and distant PPTs, no significant influence of patient age, activity level, and duration of symptoms was found. Meta-regressions based on results of the meta-analyses of HPT and CPT, were not possible due to the low number of studies reporting these outcomes.

### 3.5. Prevalence of Pain Sensitisation

The prevalence of features of pain sensitisation was evaluated in eight studies. Regarding rotator cuff tears, Karasugi et al., using painDETECT, found a prevalence of 10.9% (12/110) whereas Ko at al., using Dolour Neuropathic 4, documented a prevalence of 14.5% (16/101) [31,32]. Two studies were focused on greater trochanteric pain syndrome: Ferrer-Peña et al. found a prevalence of abnormal conditioned pain modulation of 65.3% (32/49), whereas French et al. documented a prevalence of pain sensitisation of 44.4% (8/18) using the Central Sensitization Index [26,47]. Van Wilgen et al. evaluated the prevalence of pain sensitisation in a small cohort of patients with patellar tendinopathy using Dolour Neuropathic 4, finding no sensitised subjects (0/12) [48]. Lagas et al. evaluated a cohort of patients with Achilles tendinopathy documenting a prevalence of 15% (12/80) using painDETECT. Wheeler et al. performed two studies documenting the prevalence of pain sensitisation in subjects with lower limb tendinopathies using painDETECT and central sensitisation index questionnaires: in the study using painDETECT the documented prevalence was 30.7% (23/75) in greater trochanteric pain syndrome, 0% (0/4) in quadriceps tendinopathy, 10% (1/10) in patellar tendinopathy, 26.9% (18/67) in Achilles tendinopathy, and 28.6% (36/126) in plantar fasciitis; in the study using the Central Sensitisation Index the documented prevalence was 25.9% (28/108) in greater trochanteric pain syndrome, 0% (0/12) in patellar tendinopathy, 8.6% (7/81) in Achilles tendinopathy, and 23.6% (26/110) in plantar fasciitis [45,46].

### 3.6. Methodological Quality of the Studies

Detailed results of the evaluation of the methodological quality of the included studies with the Newcastle–Ottawa scale are reported in Appendix A. Consensus between the two reviewers rating the methodological quality was reached in all evaluations. The definition of cases was considered appropriate in all the included studies since it was based on validated diagnostic criteria and excluded subjects with diseases that may influence results (i.e., neuropathies, other musculoskeletal diseases). The representativeness of the included subjects was rated as inappropriate in six studies: Fernandez del la Peñas et al., Hamstra-Wright et al., Jespersen et al., and Plinsinga et al. included only, or almost only, females [25,29,30,38], whereas Van Wilgen et al. [48] and Vallance et al. [44] included only males. The selection and the definition of controls were rated appropriate in all studies including a control group. Control groups were stratified by age and other variables in all studies except for those of Paul et al. [36] (which included older subjects and predominantly Caucasian subjects in the patients group), Plinsinga et al. [40] (which included subjects with a higher BMI in the patients group), Plinsinga et al. [38] (which included subjects with a higher BMI in the patients group), and Riel et al. [14] (which included more females and patients with a higher BMI in the patients group). The ascertainment of exposure was appropriate in all studies, and the same method was used for both groups.

## 4. Discussion

The main finding of this study is that pain sensitisation is a feature of tendinopathies, as attested by the meta-analyses on local, contralateral, and distant PPTs. However, while results indicating a lower local PPT in the affected patients were consistent among different studies, results regarding contralateral and distant pain thresholds were conflicting among different studies and tendon diseases.

PPTs, defined as the minimum level of pressure at which patients report pain when an increasing pressure is applied at a specific site with the use of an algometer, are a reliable method to assess pain sensitivity in musculoskeletal pain diseases [49]. These tests may be performed at the original site of pain or at distant uninvolved sites. The presence of a decreased pain threshold at the site of the disease (also referred as hyperalgesia) seems to be due to the sensitisation of the peripheral nociceptors (C-fibres) of deep somatic tissues [50], and it is usually referred to as peripheral sensitisation [51]. Conversely, central sensitisation implies the amplification of the sensory input with an increase in the excitability of the neurons of the dorsal horn, leading to an increased responsiveness of the central nervous system to nociceptive stimuli [51]. This process, after an initial phase characterized by the expansion of the area of pain with hyperalgesia and allodynia, leads to the spread of hypersensitivity also to distant and non-injured areas (secondary hyperalgesia) [52]. As a consequence, the evaluation of PPTs at distal and contralateral sites may be considered a diagnostic tool able to detect the involvement of the central nervous system in the enhancement of pain perception [53].

In this light, the results of the present meta-analysis document the presence of pain sensitisation in chronic tendinopathies, underlying, in particular, the contribution of peripheral pain sensitisation. In fact, while results on local PPTs clearly and consistently demonstrate an increased sensitivity at the affected site, the presence of widespread hyperalgesia was less commonly reported. Indeed, even though the overall results of the meta-analyses show the presence of an augmented contralateral and distant sensitivity, only a part of the studies was able to detect lower PPTs in the affected subjects. Notably, the results of the evaluation of contralateral and distant PPTs differed among various tendinopathies. In particular, for lateral epicondylalgia, a widespread hypersensitivity was reported in six out of seven studies and in four out of five studies evaluating contralateral and distant pain thresholds, respectively. The only study on GTPS documented a higher distant PPT in affected subjects. Results on shoulder impingement syndrome, patellar tendinopathy, Achilles tendinopathy, and plantar heel pain were more conflicting with widespread sensitivity reported in two out of five studies, one out two studies, two out of six studies, and one out of two studies, respectively.

These findings support the idea that tendinopathies may differ in terms of underlying pain processing alterations. A recent review by Rio et al. distinguished between upper limb and lower limb tendinopathies suggesting there was more consistent evidence for widespread analgesia in upper limb tendinopathies [54]. However, their results on upper limb tendinopathies were led by the results on lateral epicondylalgia (in which the influence of pain sensitization on patient pain perception is better studied), whereas the heterogenous results on lower limb tendinopathies were influenced by the different diseases considered and by the lower number of studies included. Simply distinguishing between upper and lower limb tendinopathies may thus lead to an oversimplification, merging the results of various diseases with a different etiopathogenesis that potentially influences the complex mechanism of pain sensitization [55]. Moreover, as well as possible etiopathological differences among different tendinopathies, patient characteristics might also play an important role. Mc Auliffe et al. [15] and Rio et al. [54] hypothesized that patients included in studies on upper limb tendinopathies are usually older and less active, which might influence the results of QST protocols, as a younger age and a higher activity level have been suggested to present a lower risk of developing central sensitisation [56,57]. However, while confirming the overall influence of age and activity level, this meta-regression analysis did not support the hypothesis of Mc Auliffe et al. Both age and activity level were found to correlate with differences in PPTs, but a higher difference was reported in studies including younger patients or athletes. Conversely, the meta-regression reported no influence on the results of the meta-analysis of symptoms’ duration, a factor that is often considered as important for the development of pain sensitization [58]. These controversial findings suggest a complex interplay of different factors, as the results of studies on various tendinopathies could not simply be explained by these variables. To this regard, the more common and broader abnormalities in pain processing in patients suffering from lateral epicondylalgia may be due to the characteristics of the disease itself rather than to the characteristics of the affected patients.

Among different tendinopathies, studies on epicondylalgia showed more evidence of disease features that directly support the influence of the central nervous system on perceived pain. Burns and colleagues, using surface electromyography and transcranial magnetic stimulation, demonstrated less GABA_A_- and GABA_B_-mediated intra-cortical inhibition, and less intra-cortical facilitation in the motor cortex in individuals with lateral epicondylalgia compared with healthy controls [59]. The expression of mediators involved in the development of neurogenic pain, such as glutamate [60], substance P [61], and neurokinin 1 [62], has been extensively documented in patients affected by lateral epicondylalgia, while inflammatory cells do not seems to be increased [61]. Conversely, inflammatory cells seem to play a more important role in other tendinopathies such as rotator cuff tears, patellar tendinopathy, and Achilles tendinopathy [63]. Moreover, Debenham et al. [64], although showing abnormal local two-point discrimination thresholds in subjects with Achilles tendinopathies, demonstrated similar values between the contralateral side of affected patients and unaffected site of healthy subjects, suggesting that sensitive alterations involve only the affected side in Achilles tendinopathy. Overall, the current literature findings support a higher involvement of the central nervous system in lateral epicondylalgia compared to other tendinopathies, where other mechanisms, such as local inflammatory processes, may play a stronger role in generating pain [40,65].

As well as PPT, the most complete QST protocols include the evaluation of thermal pain thresholds [66]. The data of CPTs, although included in a lower number of studies, confirmed the results of local PPT evaluations: an increased sensitivity was detected in the overall analysis. However, when the single tendinopathies were considered, only studies on lateral epicondylalgia were able to detect an increased sensitivity in the affected subjects. Regarding HPT, no difference was documented between healthy and affected subjects in the overall analysis and in all different tendinopathies. These results may be due to the fact that, whereas the mechanical stimuli is considered the best method to detect sensitisation of the deep somatic tissues’ (i.e., joint and muscle) nociceptors, thermal pain threshold may be less suitable, detecting, at best, the sensitisation of cutaneous nociceptors [51]. Moreover, a higher variability was documented in the literature for thermal pain thresholds compared to PPTs [67].

The literature presents limitations that are reflected in the present meta-analysis. The number of included studies in some of the sub-analyses is low, thus limiting the strength of these findings. Some planned analyses, such as those on distant and contralateral thermal pain threshold, temporal summation, and conditioned pain modulation, could not be performed due to lack of data. Moreover, there was a high heterogeneity in the characteristics of patients from the different trials, with a wide range of mean age (23.3 to 65.7 years) and mean pain duration (3 months to 10.6 years). A meta-regression analysis to evaluate the influence of patient-related factors was not always possible. Finally, to provide an estimate of the prevalence of pain sensitisation in chronic tendinopathies, the systematic review included also studies using questionnaires such as painDETECT, the Central Sensitization Index, and Dolour Neuropathic 4 that were initially developed to evaluate neuropathic pain and rely on questions that may reflect a broader definition of sensitivity, which includes depression, anxiety, stress, and neuroticism. However, despite the abovementioned limitations, especially pertaining the possibility to perform the planned sub-analyses, the primary analysis of the available literature offered clear results with important insights. In fact, compared to the previous literature analyses on this topic, this meta-analysis took advantage of several recent publications on pain sensitisation in tendinopathies, which allowed a quantitative synthesis on the comparison of healthy and affected subjects in terms of pain thresholds to be provided. An overall higher sensitivity was detected in patients with tendinopathies, with more evidence of pain sensitisation in patients affected by lateral epicondylalgia. Future studies should focus on the identification of pain mechanisms contributing to symptoms in different tendon diseases, in order to develop more effective treatment approaches for patients affected by tendinopathies.

## 5. Conclusions

Tendinopathies are characterized by pain sensitisation but, while features of both central and peripheral sensitisation can constantly be detected in lateral epicondylalgia, results on other tendinopathies were more conflicting, being conclusive only on the presence of peripheral sensitisation. In addition to pathophysiological differences among tendinopathies, patient characteristics are possible confounders that should be taken into account when addressing pain sensitisation.

## Figures and Tables

**Figure 1 biomedicines-10-01749-f001:**
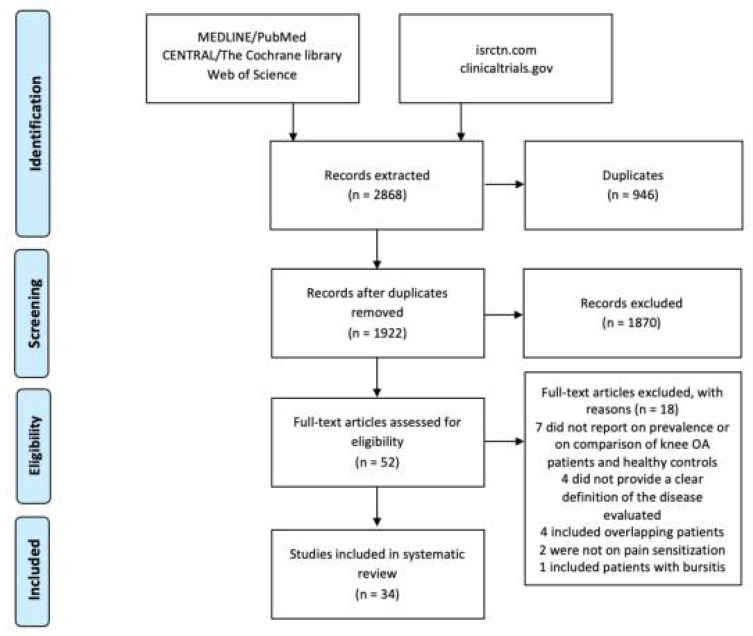
PRISMA flowchart of the study selection process.

**Figure 2 biomedicines-10-01749-f002:**
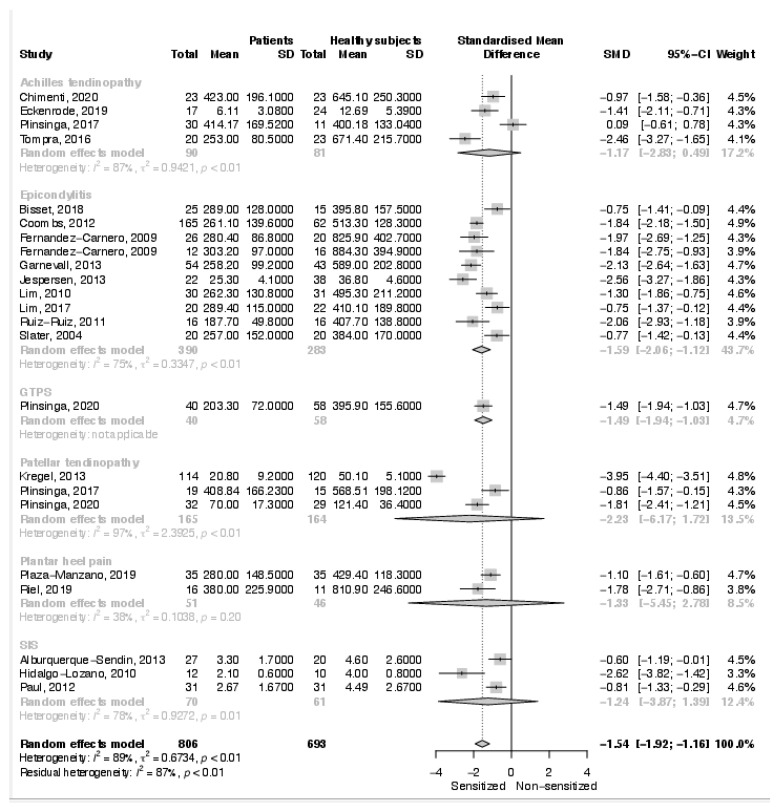
Results of the meta-analysis on local pressure pain thresholds.

**Figure 3 biomedicines-10-01749-f003:**
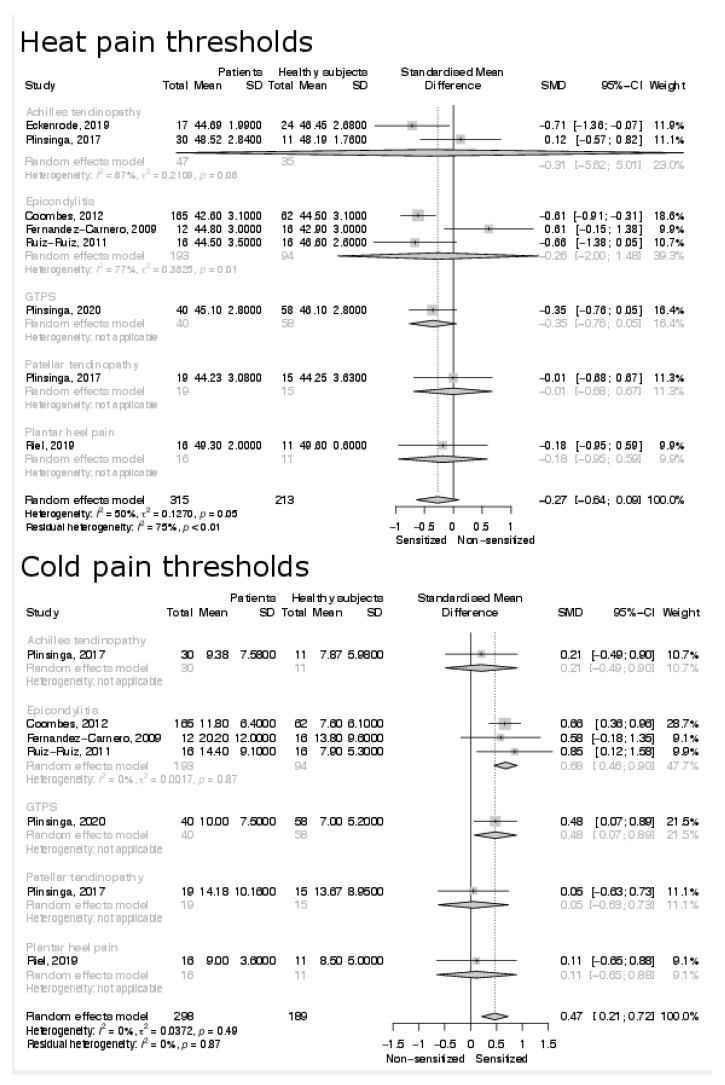
Results of the meta-analysis on local heat and cold pain thresholds.

**Figure 4 biomedicines-10-01749-f004:**
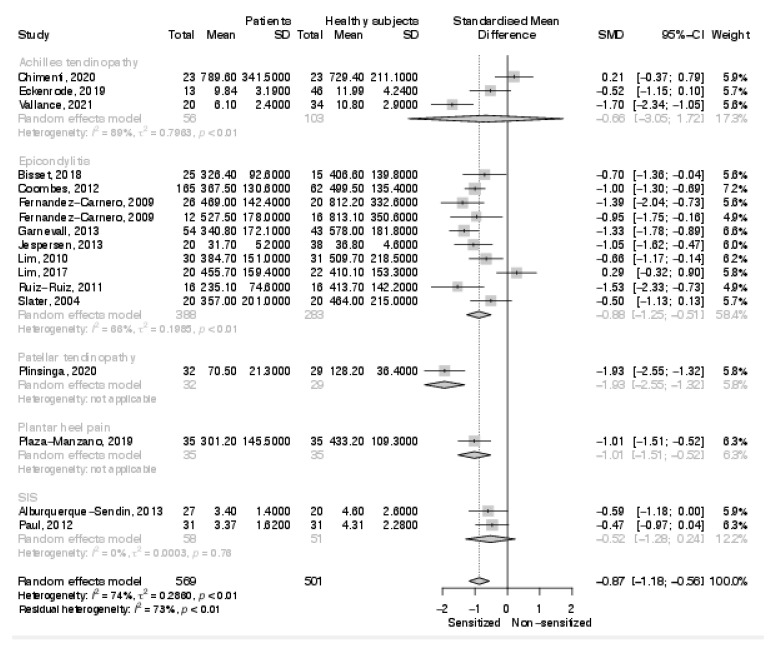
Results of the meta-analysis on contralateral pressure pain thresholds.

**Figure 5 biomedicines-10-01749-f005:**
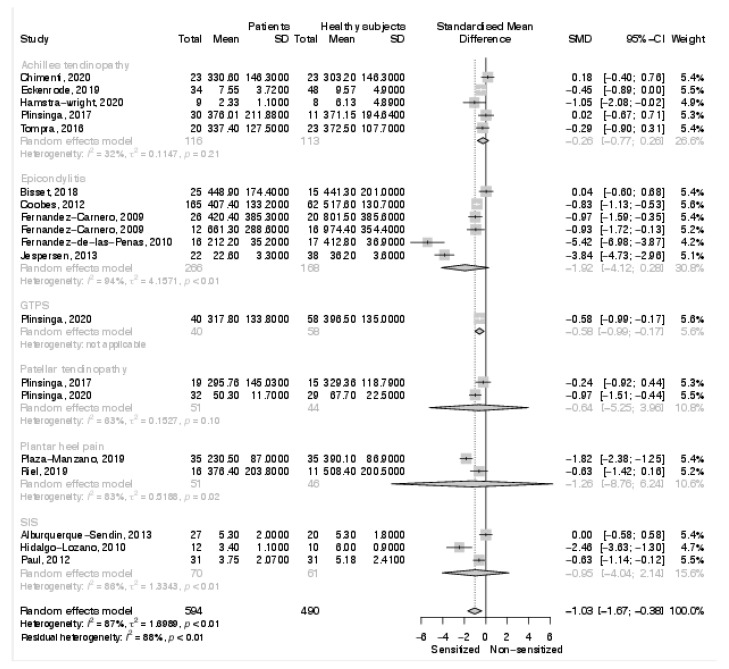
Results of the meta-analysis on distant pressure pain thresholds.

**Table 1 biomedicines-10-01749-t001:** Characteristics of the included studies and patients.

Article	Disease	Included Patients	Age	BMI	Pain Duration (Months)	Inclusion Criteria	Exclusion Criteria	Controls
**Alburquerque-Sendìn, 2013**	SIS	13M, 14F	35.6 ± 12.1 (30.8–40.4)	22.98	44.3 ± 54 (23.0–65.7)	Shoulder pain of >6 w, due to SIS; >1 + impingement test with painful ROM during elevation; or external rotation with the arm in 90° of elevation in the coronal plane	Fibromyalgia, pregnancy, a history of traumatic injury, torn tendons, ligamentous laxity, numbness/tingling in the arm, previous shoulder/neck surgery, systemic illnesses, BMI > 28, IA CS < 3 m, FKT < 6 m, depression (BDI ≥ 9), analgesics/muscle relaxants < 72 h	No upper limb disorder and matched with respect to age, weight, height
**Bisset, 2018**	LE	10M, 15F	50.4 ± 8.7	24.7 ± 3.9	3 (6–24)	Unilateral LE > 6 w. Pain over the lateral humeral epicondyle that was aggravated by palpation, gripping, and resisted wrist/finger extension	Pregnant/breast feeding, history of cardiac, systemic, or neurological disorders, other musculoskeletal pain requiring treatment < 3 m, medication affecting sensation, or history of upper limb dislocations or fractures	30–70 y (matched to LE counterparts) no history of pain lasting > 1 w or requiring treatment < 6 m, no history of LE
**Chimenti, 2020**	AT	8M, 15F	39.5 ± 10.3	33.7 ± 7.8	NR	18–70 y, English speakers, Achilles pain > 3 m, increased by activity, tenderness to palpation, stiffness after rest	Unable to manage stairs, foot/ankle surgery, pregnant/nursing, other painful diseases, adverse reaction to anesthetic	18–70 y, English speaking, no other painful diseases, (matched with AT counterparts)
**Coombes, 2012**	LE	101M, 63F	49.6 ± 9	26.5 ± 5.1	6 ± 7.5	Elbow pain over the lateral epicondyle > 6 w, aggravated by palpation, gripping and resisted wrist and/or finger extension.	Other upper limb conditions, or recent fractures, IA CS or FKT	35–70, no history of LE matched for sex. Concomitant neck or other arm pain < 6 m
**Eckenrode, 2019**	AT	8M, 9F	39 ± 10.8	NR	19.8 ± 30.3	**Active participation in regular physical activity** > 3 d/w, pain at the insertion or mid-portion of the AT > 3 m, pain with palpation to the involved AT and its insertion, pain with AT loading activities	AT tears, AT surgery, chronic pain, or inflammatory condition. Achilles region symptoms with screening tests for lumbar problems. Use of pain medications, SSRI, neurological condition, other orthopedic injury to the spine or lower extremities < 1 y, loss of sensation to the lower legs.	No musculoskeletal pain conditions < 1 y. Active participation in regular running (>5 miles/w), and no pain with a minimum of 15 single-leg active heel raises.
**Fernàndez-Carnero, 2009**	LE	10M, 16F	43 ± 10 (23–63)	NR	20.3 (11.3–29.2)	>2 of the following: pain over lateral side of the elbow, pain on palpation over the lateral epicondyle, pain on hand gripping, and pain resisted static contraction or stretching of the wrist extensor muscles. Unilateral symptoms > 2 m	Other upper extremity diseases, systemic disease; bilateral symptoms; involved with or seeking litigation; IA CS or FKT < 1 y; surgery to either elbow	History of upper extremity or neck pain, fractures or neurologic disorders, or prior strenuous wrist extensor training.
**Fernàndez-Carnero, 2009**	LE	6M, 6F	47 ± 10 (34–56)	NR	25 (10–52)	>2 of the following: unilateral pain over lateral side of the elbow > 6 w; pain on palpation over the lateral epicondyle; pain with gripping; decreased grip strength on the affected elbow; elbow pain with resisted static contraction or stretching of extensor muscles	Other upper extremity diseases, systemic disease, seeking litigation; IA CS or FKT < 1 y; surgery to either elbow	No upper extremity symptoms. Upper extremity and cervical pain < 6 m, fractures or neurological disorders, prior wrist extensor training, or analgesic or antiinflammatory drugs.
**Fernández-de-las-Peñas, 2010**	LE	16F	43 ± 7 (34–55)	NR	21.6 ± 14.4 (9.6–33.6)	>2 of the following: pain over lateral side of the elbow, pain on palpation over the lateral epicondyle, pain on hand gripping, and pain-resisted static contraction or stretching of the wrist extensor muscles. Unilateral symptoms > 2 m	Bilateral symptoms; >65 y; previous surgery or IA CS; had other diagnoses of the upper extremity; upper extremity trauma; systemic cause; pregnant; other musculoskeletal medical conditions; seeking litigation; BDI > 8.	History of upper extremity or neck pain, fractures, or neurological disorders
**Ferrer-Peña, 2018**	GTPS	8M, 41F	48.3 ± 8.1	26.6 ± 5.4	16.0 ± 17.4	Unilateral lateral hip pain, tenderness on palpation at the greater trochanter.	Hip or knee OA, presence of neurological or systemic condition, cognitive impairment or psychiatric disease or surgery or trauma at the hip, or IA CS < 6 m, bilateral hip and/or low back pain, and/or sciatica as a primary cause of pain.	/
**French, 2019**	GTPS	3M 15F	54.5 (25–76)	27	9.5	Unilateral lateral hip pain > 3 m, >18 y, pain on palpation of the greater trochanter and lateral hip pain with side lying on the affected side, during weight-bearing activities or on sitting.	Hip OA, systemic inflammatory disease, lumbar spine-related nerve root signs, spinal or ipsilateral hip surgery, neurological disease, non-English speaking or CS to the affected hip < 3 m.	No hip OA, systemic inflammatory disease, lumbar spine nerve root signs, spinal/hip surgery, neurological disease, English speaker, no CS < 3 m. No low back, hip, or groin pain
**Garnevall, 2013**	LE	16M, 38F	48.7 ± 7.5 (32–64)	NR	34.3 (1–240)	Pain on palpation of the epicondyle; pain on resisted extension of the wrist; pain on passive stretching of the wrist extensor muscles; and pain on resisted finger extension.	Previous dislocated elbow, referred cervical pain to the forearm, neurological signs, traumatic onset, CS injections < 2 m, rheumatoid arthritis, carpal tunnel syndrome.	In addition to the exclusion criteria, negative provocative tests
**Gwilym, 2010**	SIS	7M, 10F	55 (42–60)	NR	42 (9.6–240)	Unilateral shoulder pain, with a contralateral OSS ≥ 42; pain attributed to impingement; no full-thickness rotator cuff tear on HD US; no shoulder OA on X-ray; no cervical radiculopathy	NR	Free from shoulder pain with an OSS of 48
**Hamstra-Wright, 2020**	ITBS	9F	35.7 ± 11.4	NR	15	**Running** distance/week > 20 km, lateral knee pain at 30° flex during running, Noble compression test +, tenderness with palpation over lateral epicondyle or ITB	Previous knee surgery, other knee disorders, drugs affecting the outcome	**Running** distance/week > 20 km, no previous knee surgery or other knee disorders or drugs affecting the outcome
**Hidalgo-Lozano, 2010**	SIS	7M, 5F	25 ± 9 (20–38)	NR	8.5 (5–12)	Unilateral shoulder pain > 3 m and >4 on 0–10 NRS during arm elevation. Neer+ and Hawkins+ for the diagnosis of SIS	Bilateral shoulder symptoms; <18 y, >65 y; Shoulder fractures or dislocation; cervical radiculopathy; IA CS; fibromyalgia; systemic disease; shoulder or neck surgery; FKT for the neck–shoulder area < 1 y.	Age-matched, right-handed controls. No neck, shoulder, or arm pain, history of trauma, or diagnosis of any systemic disease.
**Jespersen, 2013**	LE	22F	43 ± 10.6	26.2 ± 5.9	5.6 ± 3.2	LE	Other rheumatic diseases; endocrine, cardiovascular, or pulmonary diseases; psychiatric disorders	No other rheumatic, endocrine, cardiovascular, or pulmonary diseases; psychiatric disorders, musculoskeletal pain < 1 w.
**Karasugi, 2016**	RCT	60M, 50F	65.7 ± 8.5 (46–88)	NR	9.9 ± 14.2	Shoulder pain and rotator cuff tear on MRI	Moderate/severe OA and abnormalities on X-ray, pain with cervical motion, Spurling test + or Jackson’s test +, central/peripheral nervous lesions, diabetes mellitus, shoulder surgery, duration of symptoms < 1 m or >60 m, workers’ compensation claim, NeP medication.	/
**Ko, 2018**	RCT	53M, 48F	53.2 ± 3.3 (43–59)	NR	127.2 ± 40.8 (36–160)	Required surgery for a full-thickness RCT; shoulder pain > 3 m; no trauma history; and <60 y	Shoulder surgery; bilateral; other shoulder lesion; injection therapy < 3 m; previous trauma, infection, or other inflammatory disease; possible cervical spine lesion; diabetes or neurologic disorder.	/
**Kregel, 2013**	PT	65M, 49F	23.4 ± 4.5	22.2 ± 3.6	NR	Patellar tendon pain during and after **sport.** Tenderness on palpation of the patellar tendon. A VISA-P < 80. Symptoms of PT > 3 m	Knee surgery, diabetes mellitus, neurological disease.	**Athletes** without knee pain and a VISA-P > 80
**Lagas, 2021**	AT	39M, 41F	50 (44–54)	25.7 (24.0–30.1)	16 (10–32)	18–70 y, painful swelling of the Achilles tendon, 2–7 cm proximal of the calcaneal insertion, pain > 2 m, non-responsive to >6 weeks of FKT, neovascularisations on Power Doppler US.	Achilles tendon rupture, clinical suspicion of insertional tendinopathy and inability to participate in an active exercise program	/
**Lim, 2011**	LE	21M, 9F	52 ± 9.1	24.9 ± 2.4	20.7 ± 35.3	Unilateral elbow pain > 6 w, pain over lateral epicondyle, provoked by 2 of the following: gripping, resisted wrist or middle finger extension, palpation in conjunction with reduced grip strength.	Injection < 6 w, neck or other arm pain preventing work or recreational activities participation or treated < 6 m, sources of elbow pain, pain at the radiohumeral joint, hand sensory disturbances, fractures < 10 y, elbow surgery, malignancy, inflammatory/arthritic disorder	Matched for age and sex. No history of arm or neck pain and no fractures no neurological disorders or musculoskeletal pain < 12 m.
**Lim, 2017**	LE	15M, 5F	50.7 ± 7.1	25.2 ± 4.0	10.2 ± 18.1	Unilateral elbow pain > 6 w, pain over lateral epicondyle, provoked by 2 of the following: gripping, resisted wrist or middle finger extension, palpation in conjunction with reduced grip strength.	Injection < 6 w, neck or other arm pain preventing work or recreational activities participation or treated < 6 m, sources of elbow pain, pain at the radiohumeral joint, hand sensory disturbances, fractures < 10 y, elbow surgery, malignancy, inflammatory/arthritic disorder	Matched for age and sex. No history of arm or neck pain and no fractures no neurological disorders or musculoskeletal pain < 12 m.
**Paul, 2012**	SIS	15M, 16F	51.7 ± 10	NR	NR	>21 y, shoulder pain > 6 m, and shoulder pain > 4 on 0–10 NRS	Joint or overlying skin infection, prior surgery, other chronic pain syndrome.	>21 y, without pain < 1 w > 3 on <0–10 NRS, no pain in a location > 16/30 d. No joint or overlying skin infection
**Plaza-Manzano, 2019**	Plantar heel pain	18M, 17F	41.7 ± 11	28.6 (21.8–35.4)	18.4 (11.7–25.1)	Insidious onset of sharp pain on the plantar heel surface upon weight-bearing after non-weight-bearing, increasing in the morning, pain with palpation of the proximal plantar fascia, pain for >3 months, unilateral, ≥18 y	Lower extremity surgery, ≥2 positive neurologic signs of nerve root compression, other causes of heel pain, treatment for the heel < 6 w.	Age- and sex-matched healthy controls with no lower extremity pain
**Plinsinga, 2017**	PT	14M, 5F	29.5 ± 6.6	25.4 ± 3.3	42.7 ± 39.4	Persistent pain of ≥3/10 on a 0–10 NRS > 3 m, pain and tenderness on palpation patellar tendon and Achilles’ tendon, provocation of pain on a loading test.	Injections < 12 m; previous surgery; major trauma to the knee or Achilles, any other significant musculoskeletal injuries limiting daily activities and seeking treatment < 6 m. Neurological conditions or neurological deficits, diabetes mellitus, lower back surgery or fibromyalgia	No pain or previous surgery on the patellar or Achilles’ tendon. Neurological conditions or known neurological deficits, diabetes mellitus, lower back surgery or fibromyalgia
	AT	17M, 13F	45.7 ± 11.7	28.3 ± 5.4	38.6 ± 71	Persistent pain of ≥3/10 on a 0–10 NRS > 3 m, pain and tenderness on palpation patellar tendon and Achilles’ tendon, provocation of pain on a loading test.	Injections < 12 m; previous surgery; major trauma to the knee or Achilles, any other significant musculoskeletal injuries limiting daily activities and seeking treatment < 6 m. Neurological conditions or neurological deficits, diabetes mellitus, lower back surgery or fibromyalgia	No pain or previous surgery on the patellar or Achilles’ tendon. Neurological conditions or known neurological deficits, diabetes mellitus, lower back surgery or fibromyalgia
**Plinsinga, 2020**	GTPS	2M, 38F	51 ± 9	28.5 ± 6.2	18 ± 15	18–70 y, pain > 2/10 on an 0–11 NRS, >3 m pain, pain on tendon insertion, at least 1 positive test: 30-s single-leg stance, FADER test, static muscle FADER test, FABER test, Ober’s test, static muscle contraction in the Ober’s test	Groin pain on quadrant testing > 3/10 on the 0–11 NRS, steroids < 6 m, major trauma < 12 m, or had lower limb or back pain that was worse than their hip pain, required treatment, or prevented usual activities < 6 m, Pregnancy, systemic inflammatory or neurological disorders, uncontrolled diabetes, and fibromyalgia	18–70 y, No pain preventing usual activity, no pregnancy, systemic inflammatory or neurological disorders or uncontrolled diabetes and fibromyalgia
**Plinsinga, 2020**	PT	16M, 5F	21.9 ± 2.8	23.3 ± 2.2	NR	18–65 y, **training > 1/w**, PT diagnosed, pain > 3 m, not under treatment, actually training	Previous knee or lower back surgery, bilateral pain, diabetes, neurological diseases	18–65 y, **training > 1/w**, no PT or previous knee or lower back surgery, or diabetes or neurological diseases
**Riel, 2019**	Plantar heel pain	4M, 12F	47 ± 9.4	29.3 ± 6	8.5 (6–14.5)	Plantar heel pain > 3 m; average pain intensity of ≥2 on 0–10 NRS last week; thickness of the plantar fascia > 4.0 mm on US, palpation pain of the medial calcaneal tubercle or proximal plantar fascia	>18 y; inflammatory systemic diseases; prior heel surgery; pregnancy; pain < 3 m; CS injection < 6 m other musculoskeletal injuries for which treatment was sought < 6 m	No heel pain or other lower limb pain.
**Ruiz-Ruiz, 2011**	LE	6M, 10F	45 ± 8 (32–58)	NR	19.2 ± 9.6 (13.2–34.8)	≥3 m of: pain over the lateral side of the elbow; pain on palpation over the lateral epicondyle or the associated common wrist extensor tendon; elbow pain with hand gripping; elbow pain with either resisted static contraction or stretching of the wrist extensor muscles. Unilateral symptoms > 3 m	Bilateral, >65 y, previous surgery or CS injections; other diagnoses of upper extremity; arm or neck trauma; general musculoskeletal diseases; seeking litigation, BDI-II > 8	No arm or neck pain, fractures, or any neurological disorders
**Slater, 2004**	LE	10M, 10F	48.25 (34–65)	NR	6.5 ± 4.9	Pain on palpation over the lateral epicondyle and the associated common extensor myotendinous unit; pain on functional activities, or with passive stretching of the wrist extensors. Unilateral symptoms > 3 m.	Bilateral, cervicothoracic spinal pathology, upper limb musculoskeletal or neurological disorders.	Matched for age, gender, and affected arm. No upper limb pain, fractures or neurological disorders, or prior wrist extensor training. No anticoagulant medication or medications influencing pain sensitivity
**Tompra, 2016**	AT	16M, 4F	42.9 ± 13.5	23.4 ± 2.7	21.8 ± 26.1	**Running activities** at the period of testing, activity-related pain and tenderness on tendon palpation. Pain > 3 m	Other medical condition or musculoskeletal disorder < 6 m lasting > 1 w or for which treatment was sought, systemic disorders, cardiovascular or neurological problems, fibromyalgia, and medication usage	**Runners** without Achilles tendinopathy
**Vallance, 2020**	AT	20M	45.4 ± 10.0	29.1 ± 4.6	NR	>18 y, pain to the posterior aspect of the calcaneum and at least one of: gradual onset of pain, pain aggravated by weight-bearing, and worsening after inactivity, symptoms > 3 m	Females, previous lower limb or lumbar injuries < 3 m, other painful, endocrine, or neurological diseases, bilateral AT.	>18 y, matched, males, no previous lower limb or lumbar injuries < 3 m or other painful, endocrine or neurological diseases
**van Wilgen, 2013**	PT	12M	23.3 ± 3.6	23.2 ± 3.6	30 (6–120)	Males with PT, knee pain in the proximal patellar tendon related to exercise and tenderness upon palpation of the patellar tendon. Pain > 6 m and VISA-P < 80.	Altered somatosensory function, knee surgery, diabetes, fibromyalgia, or neurological diseases	VISA-P > 90.
**Wheeler, 2017**	Lower Limb	99M, 183F	51.9 (44.0–61.8)	NR	24 (12–36)	Lower limb tendinopathy symptoms resistant to conservative management	Other causes of pain	/
**Wheeler, 2019**	Lower Limb	106M, 206F	54.9 (46.4–88.6)	NR	24 (12–36)	Chronic lower limb tendinopathy/tendon-like condition, including GTPS, patella tendinopathy, Achilles tendinopathy (both insertional and non-insertional subtypes) and plantar fasciitis. Symptoms resistant to conservative management	Other differential diagnoses	/

SIS: shoulder impingement syndrome; LE: lateral epicondylalgia; RCT: rotator cuff tear; AT: Achilles tendinopathy; GTPS: greater trochanteric pain syndrome; ITBS: Ileo-tibial-band syndrome; PT: patellar tendinopathy; M: males; F: females; BMI: body mass index; NR: non-reported; w: weeks; ROM: range of motion; d: days; m: months; y: years; OSS: Oxford Shoulder Scale; US: ultrasound; FKT: physiotherapy; NRS: Numeric Rating Scale; MRI: magnetic resonance imaging; VISA-P: Victorian Institute of Sport Assessment—Patellar; CS: corticosteroid; BDI: Beck’s depression index; NeP: neuropathic pain.

## Data Availability

Further data regarding this publication can be requested by mail to the corresponding author.

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
