# Peer review of "Tendinopathies and Pain Sensitisation: A Meta-Analysis with Meta-Regression"

_biomedicines, 2022, doi:10.3390/biomedicines10071749_

Round 1

Reviewer 1 Report

In this study the authors made their effort to investigate the evidence on pain sensitization in chronic tendinopathies.

Overall study is good and compile the information from suitable latest studies.

Introduction is well written. Rationality and objective of the study are mentioned clearly at the end of introduction part.

Although some figures have shown, high heterogeneity, however they are mentioned.  

Discussion is well written.

However, I have few minor comments as follows  

Materials and methods:

Line 75: was the date extracted on a just a single day of March 1st, 2022. Kindly mention the time frame.

Line 80: I suggest to modify the sentence as “No Institutional Review Board permission was needed to retrieve data”  

137-141: it is suggested to arrange these tendinopathies from higher number to lower number. Example 11, 5,4 and so on.

Page 153-155: Same comment as above. Arrangement

Author Response

In this study the authors made their effort to investigate the evidence on pain sensitization in chronic tendinopathies. 

Overall study is good and compile the information from suitable latest studies. 

Introduction is well written. Rationality and objective of the study are mentioned clearly at the end of introduction part.

Although some figures have shown, high heterogeneity, however they are mentioned.  

Discussion is well written. 

  • Dear reviewer, thank you for your positive feedback. Please find below the point-by-point answer to your comments.

However, I have few minor comments as follows  

Materials and methods: 

Line 75: was the date extracted on a just a single day of March 1st, 2022. Kindly mention the time frame. 

  • March 1st, 2022 is the day on which the extraction from online databases of all the citations retrieved with the string reported was performed. In the next months we performed the article selection, and the data extraction processes. Time frame are now specified as requested. (lines 83 and 95)

Line 80: I suggest to modify the sentence as “No Institutional Review Board permission was needed to retrieve data”  

  • Thank you for your suggestion. Done (line 80).

137-141: it is suggested to arrange these tendinopathies from higher number to lower number. Example 11, 5,4 and so on. 

  • Thank you, arranged as suggested (lines 139-149).

Page 153-155: Same comment as above. Arrangement

  • Thank you, arranged as suggested (lines 158-163).

Reviewer 2 Report

Thank you very much for your great effort.

This is very good paper. I have no further comments.

I assume that this study is very well done.
The paper's theme and methodology are clear, and the discussion is not a leap.
Therefore, we have no problem accepting the paper without revision.  

Author Response

Dear Reviewer, 

Thank you for your positive feedback.

Reviewer 3 Report

Dear Authors, 

your paper seems to be interested. In my opinion all of data is clearly.

Only one sugestion: 

line 76-77 you double use epicondylalgia word.

Please correct it. ;)

regards

Author Response

Dear Authors, 

your paper seems to be interested. In my opinion all of data is clearly.

Only one sugestion: 

line 76-77 you double use epicondylalgia word.

Please correct it. ;)

Dear Reviewer,

Thank you for you positive feedback. We corrected the mistake you noticed. Thank you again.